# Coordinated Expression of the Genes Encoding FocA and Pyruvate Formate-Lyase Is Important for Maintenance of Formate Homeostasis during Fermentative Growth of *Escherichia coli*

**Michelle Kammel** ® and **Robert Gary Sawers** *®

Institute for Biology/Microbiology, Martin-Luther University Halle-Wittenberg, Kurt-Mothes-Str. 3, 06120 Halle (Saale), Germany

* Correspondence: gary.sawers@mikrobiologie.uni-halle.de

**Abstract:** FocA is a pentameric membrane channel that translocates formic acid bidirectionally across the cytoplasmic membrane of *Escherichia coli* during fermentation. The *focA* gene is co-transcribed with *pflB*, which encodes pyruvate formate-lyase, the enzyme that generates formate. Recent evidence has suggested that FocA serves to regulate intracellular formate levels and thus helps to maintain pH balance in fermenting cells. In this study, we aimed to provide support for this hypothesis by either altering FocA levels, mutating the chromosomal *focA* gene, or introducing additional copies of *focA*, either alone or with *pflB*, on a plasmid and monitoring the effect on intracellular and extracellular formate levels. Our results revealed that the expression of the native *focA-pflB* operon ensures that intracellular formate levels remain relatively constant during exponential phase growth, even when additional, mutated copies of *focA* that encode FocA variants are introduced *in trans*. Enhancing *focA* expression was balanced by higher formate excretion from the cell. Using chromosomal *focA* gene variants confirmed that FocA, and not PflB, sets intracellular formate homeostatic levels. More-over, any chromosomal *focA* mutation that altered the formate concentration inside the cell caused a negative fermentative growth phenotype. Thus, FocA governs intracellular formate levels to ensure optimal growth during glucose fermentation.

**Keywords:** fermentation; formate hydrogenlyase; FocA channel; formic acid; pyruvate formate-lyase

## 1. Introduction

Enterobacterial mixed-acid fermentation is characterized by the production of varying amounts of acetic, succinic, lactic, and formic acid, as well as ethanol and the gases $H_2$ and $CO_2$ [1]. The consequence of this fermentation in a closed, batch-culture system is a progressive decrease in the pH of the culture medium, which ultimately results in the cessation of growth [2]. Attenuation of pH-restricted growth can be achieved if one or more of these acids are removed, resulting in partial deacidification of the cellular environment. Of the four acidic fermentation products, only formic acid is re-imported to any significant extent by stationary-phase cells. Upon uptake, formic acid is disproportionated to $H_2$ and $CO_2$ by the cytoplasmically oriented and membrane-associated formate hydrogenlyase complex (FHL-1) [3–6]. Notably, FHL complexes are phylogenetically related to complex I of the respiratory chain and, consequently, they have been proposed to contribute to energy conservation [6,7].

When *Escherichia coli* ferments with glucose as the energy source, up to one-third of the carbon can be converted to formate by the CoA-dependent cleavage of pyruvate by the pyruvate formate-lyase (PflB) enzyme [1,8]. Acetyl-CoA is further converted to a mixture of acetate and ethanol; typically, only relatively minor amounts of lactate and succinate are generated [1,9,10]. Re-import of formate (or formic acid) only occurs when

an active FHL-1 complex is present [10]. The uptake of the anion/acid is greatly aided by the pentameric FocA channel [3,11] which controls the distribution of formate/formic acid across the cytoplasmic membrane [3]. The results of recent studies suggest that FocA translocates formic acid (formate plus a proton) out of the cell during exponential phase growth of *E. coli* [3,12,13] and re-imports it when the pH of the medium decreases [14]. The latter correlates with the long-known phenomenon of acid pH-dependent $H_2$ production by stationary-phase *E. coli* cells [15,16].

　　In the efflux direction, FocA appears to function as a channel for formic acid, while in the uptake direction it uses a different mechanism [3]. This conclusion is based mainly on the analysis of FocA variants that have different impacts on the efficiency and direction of formate translocation. There are two highly conserved amino acid residues, threonine-91 (T91) and histidine-209 (H209) that are located centrally in the translocation pore of each protomer of the FocA pentamer [3,11,17,18]. Construction and analysis of different variants with exchanges in T91 revealed that it is crucial for the efficient efflux of formic acid, while H209 is essential for the uptake of the anion/acid [13,14,19]. Both residues interact through hydrogen-bonding contacts and the imidazolium cation at low pH has been proposed to be particularly important for the uptake and protonation of formate to allow its passage across the hydrophobic pore of FocA [3]. Using different mechanisms for efflux and uptake of formate/formic acid ($pK_a$ = 3.75), the latter being pH-dependent, should provide *E. coli* with the capability to regulate the pH of its cytoplasm, periplasm, and immediate environment outside the cell. Evidence that the bacterium can indeed do this, particularly during the stationary phase, has been provided [20], and enhanced uptake to relieve stationary phase-induced pH stress correlates with enhanced $H_2$ gas production under acidic conditions [2,21].

　　As well as being the substrate for the FHL-1 complex [1,4,5], formate is also required for the synthesis of FHL-1 [22,23]. At a threshold concentration of 5 mM [24], formate stimulates the FhlA-dependent control of expression of the *fdhF* and *hyc* genes, which encode the structural components of FHL-1 [5,25]. Completing the regulatory circuit that controls formate metabolism is PflB, which generates the signal molecule for the transcription factor FhlA. Not only does PflB generate formate, but it also controls its translocation through FocA by acting as a 'gating' protein that determines when the pore of FocA opens and closes [12,26]. Furthermore, the genes encoding FocA and PflB are co-transcribed in the *focA-pflB* operon [27,28], which provides yet another level of coordinated control of FocA and PflB synthesis, e.g., when the GTG translation initiation codon of *focA* is converted to an ATG. This results in 10-fold increased levels of FocA and, through polarity effects, 3-fold increased PflB levels [9]. Thus, intracellular formate levels are controlled by a complex regulatory protein network that includes PflB, FocA, and FHL-1, all with the express function of controlling intracellular and extracellular pH; they potentially also contribute to energy conservation (reviewed in [3]). A recent study has provided evidence supporting the notion that intracellular formate levels remain relatively constant during growth of a wild-type *E. coli* strain [14]. In contrast, a *focA* mutant (strain DH701 in the current study), which has two adjacent stop codons within the gene, which showed no significant to downstream polarity effects on *pflB* expression [9], showed perturbation of intracellular formate levels, especially during exponential phase growth [14]. Together, these findings suggest that maintaining formate at a constant intracellular level might be important, not only during the stationary phase but also during growth. Moreover, the data also suggest that the co-expression of *focA* and *pflB* as an operon might be particularly important in this regard. Here, we examined the effects on intracellular formate levels of over-producing different FocA variants in the wild-type strain and in strains synthesizing different amounts of chromosomally encoded FocA. Our findings revealed that as long as a native *focA-pflB* operon is present on the genome, or is provided on a plasmid, and regardless of the expression level of that operon, *E. coli* balances intracellular with extracellular formate levels through FocA to maintain a relatively constant level of intracellular formate. Moreover, we

revealed that *focA* mutants showed a growth deficit in early exponential phase cells, which confirms the central role of FocA in maintaining formate homeostasis.

## 2. Materials and Methods

### 2.1. Strains and Growth Conditions

The isogenic strains and the plasmids used in this study are listed in Table 1. All bacterial strains used in this study carry a chromosomal copy of l*fdhF_P*::*lacZ* [29]. Only strain DH702 has a genomic modification of the *focA* translation initiation codon. All plasmids carrying *focA* genes have the native GTG initiation codon.

**Table 1.** Strains and plasmids used in this study.

| Strains and Plasmids | Relevant Genotype or Characteristics | Reference or Source |
|---|---|---|
| Strains | | |
| MC4100 | F$^-$ *araD* Δ(*argF lac*) U 169 *ptsF25 deoC1 relA1 fblB530 rpsL 150* λ$^-$ | [30] |
| DH4100 | MC4100 λ(*fdhF*::*lacZ*)—parental strain | [29] |
| DH701 | REK701 λ(*fdhF*::*lacZ*)—has two stop codons at codon positions 114 and 115 in the *focA* gene; the *focA* gene is present but the strain cannot make FocA | [29] |
| DH702 | REK703 λ(*fdhF*::*lacZ*)—chromosomal GTG→ATG exchange in *focA*; overproduces FocA | [13] |
| DH4200 | MC4200 λ(*fdhF*::*lacZ*—has a chromosomal exchange at codon 209 (His→Asn) in the *focA* gene; FocA$_{H209N}$ has an efflux-only phenotype | [13] |
| DH4300 | MC4300 λ(*fdhF*::*lacZ*)—chromosomal exchange at codon 91 (Thr→Ala) in the *focA* gene; FocA$_{T91A}$ has an efflux-defective phenotype | [14] |
| Plasmids | | |
| pfocA | Amp$^r$, expression vector with the gene *focA* (without StrepII tag) | [13] |
| pfocA-T91A | Similar to pfocA, but codon threonine 91 exchanged for alanine | [19] |
| pfocA-H209D | Similar to pfocA, but codon histidine 209 exchanged for aspartic acid | [19] |
| pfocA-H209N | Similar to pfocA, but codon histidine 209 exchanged for asparagine | [13] |
| p29 | Cm$^r$, *focA*$^+$ *pflB*$^+$ *pflA*$^+$ | [27,31] |

Anaerobic growth of all strains was in M9-minimal medium, pH 7.0 [32], including 0.8 % (*w/v*) glucose as a carbon source [13]. When required, ampicillin was added to a final concentration of 150 μg mL$^{-1}$, kanamycin to 100 μg mL$^{-1}$, and chloramphenicol to 25 μg mL$^{-1}$. The *focA* gene was cloned in pASK-IBA3 vector behind the *tet* promoter; however, induction of promoter activity by supplementation of the growth medium anhydrotetracycline was not necessary to achieve gene expression [13]. Cultivation of cells at 37 °C was performed in either 130 mL sealed serum bottles, when samples were taken hourly, or Hungate tubes, when translocation activity was investigated during the exponential growth phase (OD$_{600}$ ~0.7–0.85). The samples were taken as indicated in the figures for

the determination of β-galactosidase enzyme activity or the measurement of extracellular formate levels.

### 2.2. Determination of β-Galactosidase Enzyme Activity

The β-galactosidase enzyme activity of the harvested cells was determined and calculated as reported previously [12,33]. All experiments were performed at least in duplicate with minimally three biological replicates and the values determined are presented with the standard deviation of the mean.

### 2.3. Measurement of Extracellular Formate Concentration

Analysis of extracellular concentrations of formate/formic acid and other organic acids was done exactly as described [12]. Measurements were carried out at least twice, each time with minimally three biological replicates, and the values determined are presented with the standard deviation of the mean.

## 3. Results

### 3.1. FocA Offers a Growth Advantage during Early Exponential Phase Growth

While our previous analysis of changes in intracellular formate levels during batch-culture fermentative growth of *E. coli* revealed that mutants lacking FocA accumulated formate intracellularly, when compared with the wild-type strain, no clear growth phenotype at pH 7 was observed, especially considering the final optical density of the culture attained in the stationary phase [14]. It was noted, however, that early exponential phase cultures of strains DH701 (lacks FocA) and DH4300 (synthesizes FocA$_{T91A}$) began accumulating formate intracellularly more quickly than the wild-type. Therefore, to examine growth in greater detail at the beginning of the fermentation, we carefully measured optical density, changes in intracellular formate concentration (determined by measuring β-galactosidase enzyme activity of the formate-responsive *fdhF$_P$*::*lacZ* reporter), and determined extracellular formate concentration in the medium during the first 4 h of growth (Figure 1). While β-galactosidase enzyme activity for DH4100 (parental; compare Table 1) was first detectable only after 2 h of growth (Figure 1a), increased intracellular formate levels were detectable within 1 h in both the *focA* mutant DH701 (Figure 1b) and in strain DH4300 (Figure 1c). After 4 h of growth, the parental strain, DH4100, registered a low level of formate inside the cell, while strains DH701 and DH4300 showed β-galactosidase activities that were three- and seven-fold higher, respectively, than that detected in the parent strain. Measurement of formate levels in the medium after 4 h of growth revealed that mutants DH701 and DH4300 excreted approximately 0.5 mM formate, while the parental strain DH4100 excreted approximately 2.5-fold more formate (Figure 2a). When the concentration of excreted acetate in the growth medium was determined, there was not a significant difference between the three strains (Figure 2b). Together, these results indicate that native FocA efficiently and specifically excretes formate during the early phase of fermentative growth. In contrast, a strain either lacking FocA (DH701), or a strain synthesizing a FocA variant that cannot efficiently translocate formate out of the cell (DH4300; [14]), accumulate formate intracellularly, essentially from the beginning of fermentative growth.

Determination of the growth rate of the parental strain, DH4100, (Figure 1a) showed it to have a $\mu = 0.56 \pm 0.02$, while that of DH701 ($\mu = 0.39 \pm 0.01$) and that of DH4300 ($\mu = 0.41$) were, in comparison, reduced by approximately 30%. This represents the first clearly defined growth phenotype determined for a *focA* mutant grown at pH 7 under fermentative conditions with glucose as a carbon source.

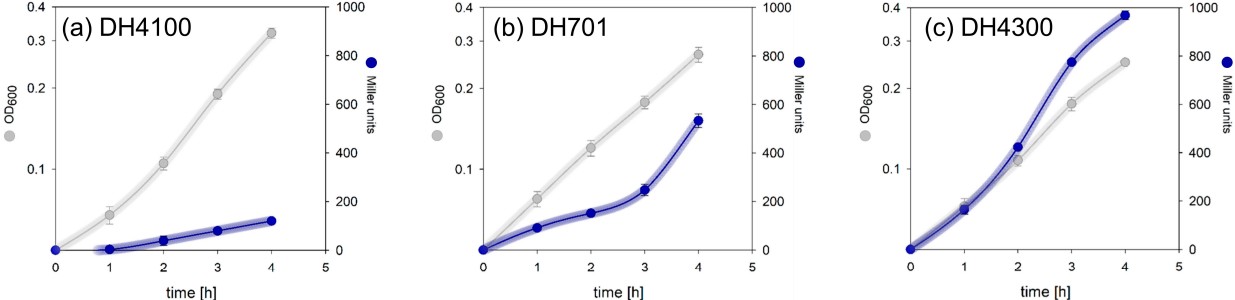

**Figure 1.** Impaired formate homeostasis and growth rate of *focA* mutants. Shown are the optical densities (gray lines) and formate-reporter enzyme activities (blue lines), representing changes in intracellular formate levels, for the indicated strains during the first 4 h of fermentative growth. Strains: (**a**), DH4100, parental; (**b**), DH701 (*focA*); and (**c**), DH4300, synthesizes $FocA_{T91A}$.

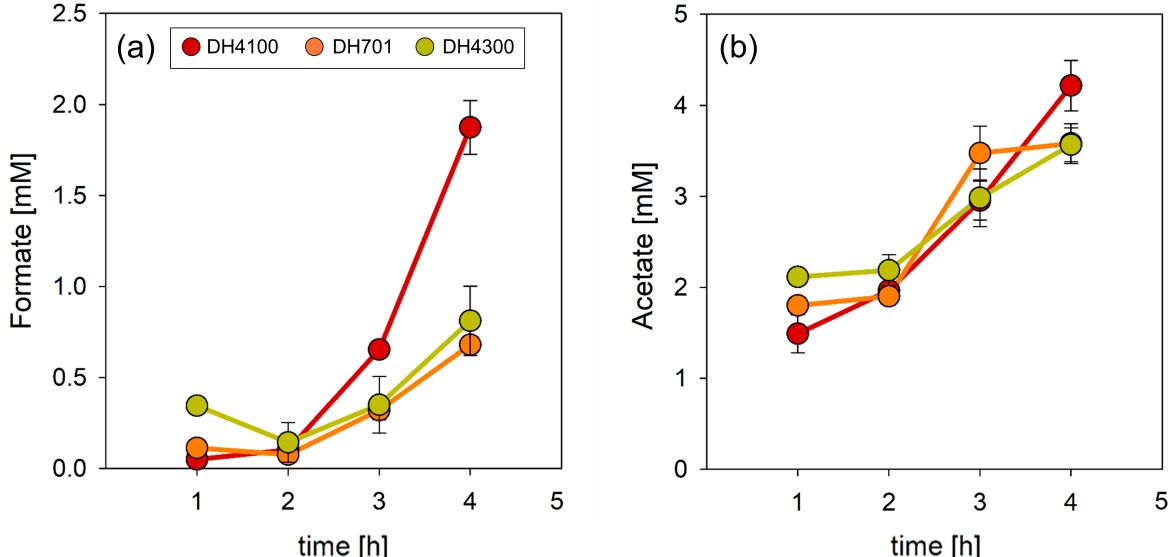

**Figure 2.** Extracellular formate and acetate concentrations. The formate (**a**) and acetate (**b**) concentrations in the minimal medium during the first 4 h of growth of the indicated strains (see key within the figure) are shown. Strains: DH4100 (red), parental; DH701 (orange), (*focA*); and DH4300 (pale green), synthesizes $FocA_{T91A}$.

### 3.2. Increasing the Copy Number of focA Does Not Affect Intracellular Formate Levels in the Parental Strain

Cultures of DH4100 grown anaerobically to the mid-exponential phase of growth ($OD_{600}$ = 0.6–0.7) had a β-galactosidase enzyme activity of ~370 units (Figure 3a). Measurement of extracellular formate levels in these cultures delivered a concentration of $4.4 \pm 0.4$ mM $OD_{600}^{-1}$ (Figure 3b). Introduction of the native *focA* gene on plasmid pfocA [12] resulted in an approximate 5% increase in β-galactosidase activity (Figure 3a). Relative to the parental strain, DH4100, which synthesizes chromosomally encoded FocA, expression of multicopy *focA* from plasmid pfocA resulted in a minimally 10- to 20-fold increased level of membrane-associated FocA in the cell [10,34]. Nevertheless, this substantial increase in FocA had no apparent impact on the intracellular formate level in DH4100 when compared with the level attained if only native, chromosomally encoded FocA was present in the cell (Figure 3a). However, elevated FocA levels resulted in an increase in the extracellular formate level of 50% compared to the parental strain without pfocA (Figure 3b). As a control, the introduction of pfocA into the *focA* mutant, DH701, reduced β-galactosidase enzyme activity by 55% (to $370 \pm 8$ units) compared to the mutant without the plasmid (Figure 3a). The introduction of pfocA had no effect on extracellular levels of

formate compared with strain DH701 (Figure 3b), which agrees with previous findings that extracellular formate levels in late-exponentially growing cells were similar between the parental strain and the *focA* mutant [14]. The restoration of parental intracellular formate levels to the *focA* mutant by native FocA confirmed our previous findings [13] and further demonstrated the regulatory control exerted by FocA on formate metabolism. Moreover, these findings verified that the plasmid-encoded gene product was functional during the current experiments.

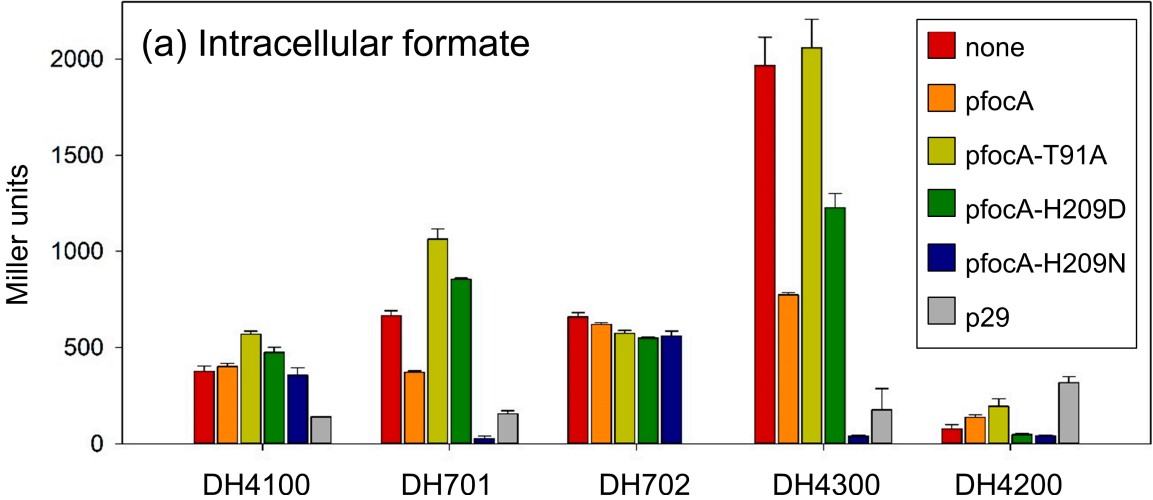

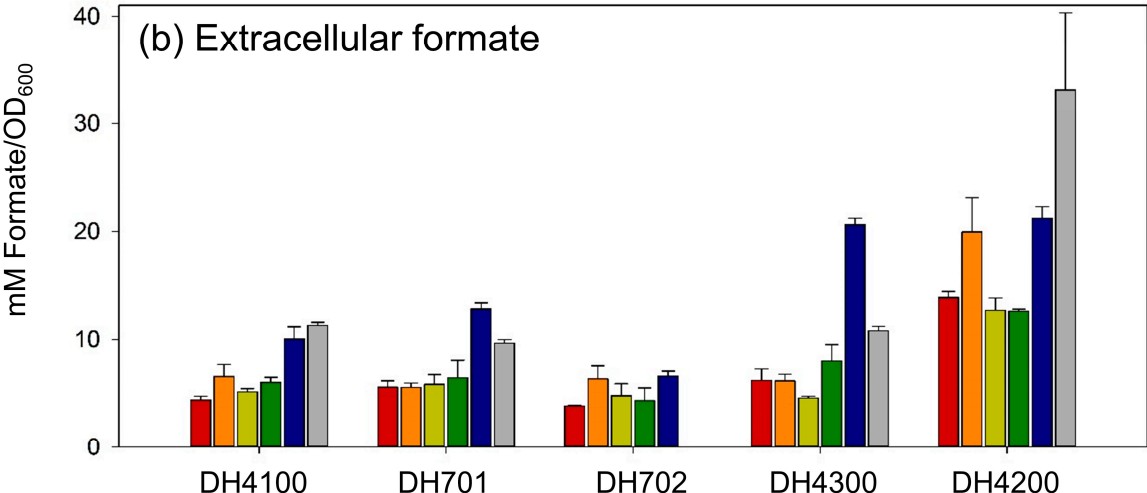

**Figure 3.** Strains with a genomic copy of the native *focA-pflB* operon maintain intracellular formate homeostasis during fermentative growth. Changes in intracellular (**a**) and extracellular (**b**) formate levels were monitored as indicated in the methods section. The strains indicated on the x-axis included the parental wild-type strain DH4100, the *focA* mutant DH701, strain DH702 with the translation initiation codon of the chromosomal *focA* gene changed to ATG, strain DH4300 encoding FocA$_{T91A}$, and strain DH4200 encoding FocA$_{H209N}$. Due to the introduction of the more efficient ATG translation initiation codon into the *focA* gene in strain DH702, 10-fold more FocA is synthesized in these cells [9]. Each strain was transformed with either no plasmid (red), pfocA (orange), pfocA-T91A (pale green), pfocA-H209D (dark green), pfocA-H209N (dark blue), or p29 (grey). Note that plasmid p29 was not tested in strain DH702.

Plasmid pfocA-H209N carries a *focA* gene encoding FocA$_{H209N}$, which functions very efficiently to excrete formate from the cell [13]. When introduced into the *focA* mutant

DH701, the β-galactosidase enzyme activity in the mutant was reduced approximately 15-fold compared with the level in DH4100 (Figure 3a). In contrast, the external formate concentration in cultures of DH701/pfocA-H209N was almost three-fold higher compared with cultures of the wild-type, parental strain (Figure 3b). Next, we tested the effect of introducing plasmid pfocA-H209N into the parental strain, DH4100. Surprisingly, the presence of this plasmid did not markedly alter intracellular formate levels when compared with the strain synthesizing only chromosomally encoded, native FocA (Figure 3a). Nevertheless, the concentration of formate measured in the medium after growth of strain DH4100/pfocA-H209N was double that of the same strain without a plasmid (Figure 3b). This result indicates that $FocA_{H209N}$ was functional in excreting formate efficiently when introduced into DH4100.

Two further FocA variants that both show defects in excretion of formate were tested next. $FocA_{T91A}$ and $FocA_{H209D}$ have been shown to have severe deficiencies in the efflux of formate from fermenting *E. coli* cells [14,19]. As a control, verification of this intracellular formate accumulation was shown for both plasmid-encoded FocA variants upon their introduction into strain DH701 (*focA*), whereby $FocA_{T91A}$ and $FocA_{H209D}$ resulted in β-galactosidase enzyme activities that were 60% and 28% higher, respectively, than that measured for strain DH701 without a plasmid (Figure 3a).

Introduction of either pfocA-T91A or pfocA-H209D into the DH4100 parental strain resulted in a comparable increase in β-galactosidase enzyme activity to that in the DH701 (*focA*) background, to between 475–570 U (Figure 3a). This result again suggests that the presence of the chromosomally encoded native FocA was capable of overcoming to a great extent the efflux-deficiency of the more abundant FocA variants [14,19].

### 3.3. Increased Synthesis of Chromosomally Encoded Native FocA Does Not Significantly Affect the Intracellular Formate Concentration

The translation initiation codon of the *focA* transcript is GUG and conversion of this codon to an AUG improves translational efficiency such that the resultant strain, REK702 (similar to DH702 but lacking λ*fdhF_P*::*lacZ*, Table 1), synthesizes approximately 10-fold more FocA than the parental strain, DH4100 [9,10]. Due to polarity effects on the downstream *pflB* co-transcript, strain DH702 also exhibits a two- to three-fold increased level of PflB [9,10]. Measurement of β-galactosidase enzyme activity after anaerobic growth of strain DH702 in M9-glucose minimal medium yielded 660 units, which corresponds to a 75% increase in β-galactosidase enzyme activity in this strain compared with that in the parental strain, DH4100 (Figure 3a). The extracellular formate concentration in the growth medium was determined to be approximately 4 mM $OD_{600}^{-1}$ (Figure 3b). After introduction of plasmid pfocA into strain DH702 and subsequent anaerobic growth, the intracellular formate level did not change significantly when compared with that of DH702 without the plasmid (Figure 3a), while the extracellular formate concentration increased by 65% (Figure 3b). In a similar experiment carried out with strain DH702 transformed with plasmid pfocA-H209N, a β-galactosidase enzyme activity of 560 units was measured (Figure 3a), while the external formate concentration increased marginally to 6.6 mM $OD_{600}^{-1}$ (Figure 3b). Surprisingly, even after introduction of plasmids pfocA-H209D or pfocA-T91A into strain DH702, intracellular formate levels were maintained at a relatively constant level of 550–575 units, with approximately 4.6 mM $OD_{600}^{-1}$ formate measured extracellularly (Figure 3a,b).

### 3.4. Co-Expression of focA and pflB Is Important in Determining Formate Homeostatic Levels

The chromosomal *focA* gene in strain DH4300 encodes the $FocA_{T91A}$ variant, which fails to translocate formate efficiently out of the cell [14]. Consequently, after anaerobic growth of strain DH4300 in M9-glucose minimal medium, the β-galactosidase enzyme activity was approximately five times higher than the level measured in the parental strain, DH4100, after growth under the same conditions (Figure 3a). Introduction of the native *focA* gene on plasmid pfocA into strain DH4300 caused a 2.5-fold reduction in β-galactosidase

enzyme activity; however, this enzyme activity was still two-fold higher than that measured in DH4100. As anticipated, the introduction of pfocA-T91A into strain DH4300 failed to lower the intracellular formate concentration (Figure 3a). In contrast, however, the introduction of plasmid pfocA-H209N into the strain caused a 50-fold reduction in β-galactosidase enzyme activity (Figure 3a), corresponding to a greatly decreased intracellular formate concentration. Concomitantly, synthesis of $FocA_{H209N}$ in the strain resulted in the accumulation of nearly 21 mM $OD_{600}^{-1}$ formate in the growth medium, a greater than three-fold increase compared with strain DH4300 without a plasmid (Figure 3b).

Strain DH4200 has a chromosomal copy of *focA* in which codon 209 decodes as asparagine and not as histidine [13]. Consequently, because $FocA_{H209N}$ has increased efflux efficiency, the strain has a low intracellular formate concentration which is reflected in a low β-galactosidase enzyme activity of $78 \pm 21$ units and is five-fold lower than in the parental strain DH4100 (Figure 3a). In contrast to the 4.4 mM formate $OD_{600}^{-1}$ excreted into the growth medium by the parental strain, strain DH4200 accumulated nearly 14 mM formate $OD_{600}^{-1}$ in the growth medium (Figure 3b). Surprisingly, the introduction of the native *focA* gene on pfocA into DH4200 only increased the β-galactosidase enzyme activity 1.8-fold compared with the plasmid-free strain (Figure 3a), while extracellular formate levels were increased further to $20 \pm 3.2$ mM $OD_{600}^{-1}$ (Figure 3b). Even the introduction of plasmid pfocA-T91A resulted in only a 2.5-fold increase in β-galactosidase enzyme activity compared with DH4200 without a plasmid (Figure 3a). This activity was still 50% lower than the enzyme activity measured for the parental strain, DH4100. Together, these results suggest that the co-expression of the *focA* and *pflB* genes in the *E. coli* genome is crucial in allowing fine-tuning of the intracellular formate concentration and that FocA, not PflB, determines the level of the anion in the cell.

To provide further support for the suggested importance of *focA-pflB* co-expression in controlling intracellular formate levels, we tested the impact of introducing the plasmid-borne *focA-pflB* operon on intra- and extracellular formate levels in the strains. Plasmid p29 (Table 1) carries the complete *focA-pflB* operon, including its regulatory region, as well as the *pflA* gene, which encodes the PflB-activating enzyme [31]. The plasmid results in an approximate 15-fold increase in the copy-number of the genes and their products [35]. The introduction of p29 into the parental strain, DH4100, caused an approximate five-fold reduction in the β-galactosidase enzyme activity compared with DH4100 without a plasmid (Figure 3a). A concomitant ~2.5-fold increase in the concentration of excreted formate was measured (Figure 3b). Similar results were determined for the *focA* mutant, DH701, when it was transformed with plasmid p29 (Figure 3a,b). The introduction of the *focA-pflB* operon on plasmid p29 into strains DH4300 (synthesizing chromosomally encoded $FocA_{T91A}$) and DH4200 (synthesizing chromosomally encoded $FocA_{H209N}$) resulted in β-galactosidase enzyme activities of approximately 320 units and 180 units, respectively, representing a 6.6-fold reduction and 4-fold increase in intracellular formate levels compared with the respective plasmid-free strains (Figure 3a). The introduction of p29 caused an increase in extracellular formate levels by 66% in strain DH4300 (to 10 mM $OD_{600}^{-1}$) and resulted in a 50% increase in extracellular formate levels (to 30 mM $OD_{600}^{-1}$) in strain DH4200 (Figure 3b). Together, these results support the proposal that co-regulated expression of the *focA* and *pflB* genes plays an important role in determining intracellular formate levels in fermenting *E. coli* cells.

## 4. Discussion

The data presented in this study provide further evidence supporting the previously suggested role of FocA in maintaining formate homeostasis in fermenting *E. coli* cells [14]. Here, we tested the impact of introducing additional copies of the *focA* gene on a plasmid (in trans) into the parental strain DH4100, or of improving the translational efficiency of the *focA* mRNA (in cis), and consequently up-regulating FocA synthesis, on intracellular formate levels. Regardless of which approach was taken to increase the dosage of native FocA molecules in the cell, the overall level of formate remained relatively constant; this

was evident for both the parental strain DH4100 and the FocA-overproducing strain DH702. Introducing plasmids carrying genes that encode amino acid variants of FocA, which were either impaired in formate efflux (e.g., $FocA_{T91A}$ or $FocA_{H209D}$), or exhibited enhanced efficiency of efflux capability (e.g., $FocA_{H209N}$), surprisingly also did not significantly affect the intracellular formate levels. However, this was only observed for a strain bearing a genomic copy of the *focA-pflB* operon, encoding native FocA and PflB, and regardless of whether increased levels of native FocA were achieved by substituting the GTG for an ATG translation initiation codon in the chromosomal *focA* gene in strain DH702. Moreover, even though the introduction of plasmid pfocA-H209N (encoding $FocA_{H209N}$) into either of these strains did not affect the intracellular formate level, it caused significantly increased levels of formate excretion. This underscores our proposal that the role of FocA is to maintain a relatively constant intracellular level of formate during exponential growth.

However, if the dosage of the *focA-pflB* operon was increased by introducing it on a plasmid into the parental strain, this caused enhanced formate efflux and a lower overall intracellular formate level. The same results were observed when the *focA* mutant, DH701, was transformed with the same plasmid. This result indicates that the effect of introducing multiple copies of the *focA-pflB* operon is independent of the genomic *focA-pflB* operon and suggests that when both FocA and PflB are co-overproduced, the equilibrium between intracellular and extracellular formate levels is shifted in favor of the extracellular environment. This was not observed when FocA was over-produced on its own, supporting a role for PflB in controlling FocA-dependent formate levels. The demonstration that PflB's interaction with the *N*-terminal domain of FocA is crucial in controlling formate translocation into and out of the cell [12] might be the reason for this observed interdependence between the two proteins. Moreover, this may also explain the exquisitely complex transcriptional and translational regulation of the *focA-pflB* operon demonstrated in early regulatory studies of anaerobic metabolism in *E. coli* [9,27,28] and highlights the importance of maintaining the co-synthesis of FocA and PflB, as well as the activity of PflB, to afford the cell optimal control of fermentative metabolism [1,5,9].

Why is it necessary to maintain a threshold level of formate in the cytoplasm? Cytoplasmic formate levels are maintained by five key factors: the generation of formate by PflB; sensing of formate by FhlA; formate-FhlA-dependent regulation of FHL-1 synthesis; disproportionation of formate by FHL-1; and efflux–influx of formate ($+H^+$) across the membrane by FocA (see Figure 4). Further levels of fine-tuning control not considered in detail here include allosteric regulation of PflB enzyme activity [8], HycA-dependent feedback control of the formate-FhlA interaction [24,25], and both the cytoplasmic and periplasmic enzymic oxidation of formate by the cell's three formate dehydrogenases [3–5]; however, periplasmic oxidation of formate is dependent on the availability of exogenous or endogenous electron acceptors [4,5]. It is proposed that in wild-type cells, the intracellular formate concentration is mainly maintained by the relative efflux–influx efficiencies of FocA, the activity of PflB, and is further fine-tuned by the disproportionation of excess formate to $H_2$ and $CO_2$ by the active FHL-1 complex (Figure 4). Maintenance of this homeostatic equilibrium of formate is consequently perturbed if the functions of FocA, PflB, or the FHL-1 complex are impaired or abolished. This is exemplified by *E. coli* mutants unable to synthesize an active FHL-1 complex, which fail to reimport formate once it has been excreted into the periplasm or growth medium and concomitantly the mutants show impaired growth [10]. Moreover, *pflB* mutants fail to synthesize FHL-1 due to lack of inducer formate, unless the anion is provided exogenously [23]. The *pflB* mutants also exhibit aberrant formate uptake kinetics because PflB is unavailable to facilitate controlled FocA-dependent uptake of formate [12,26].

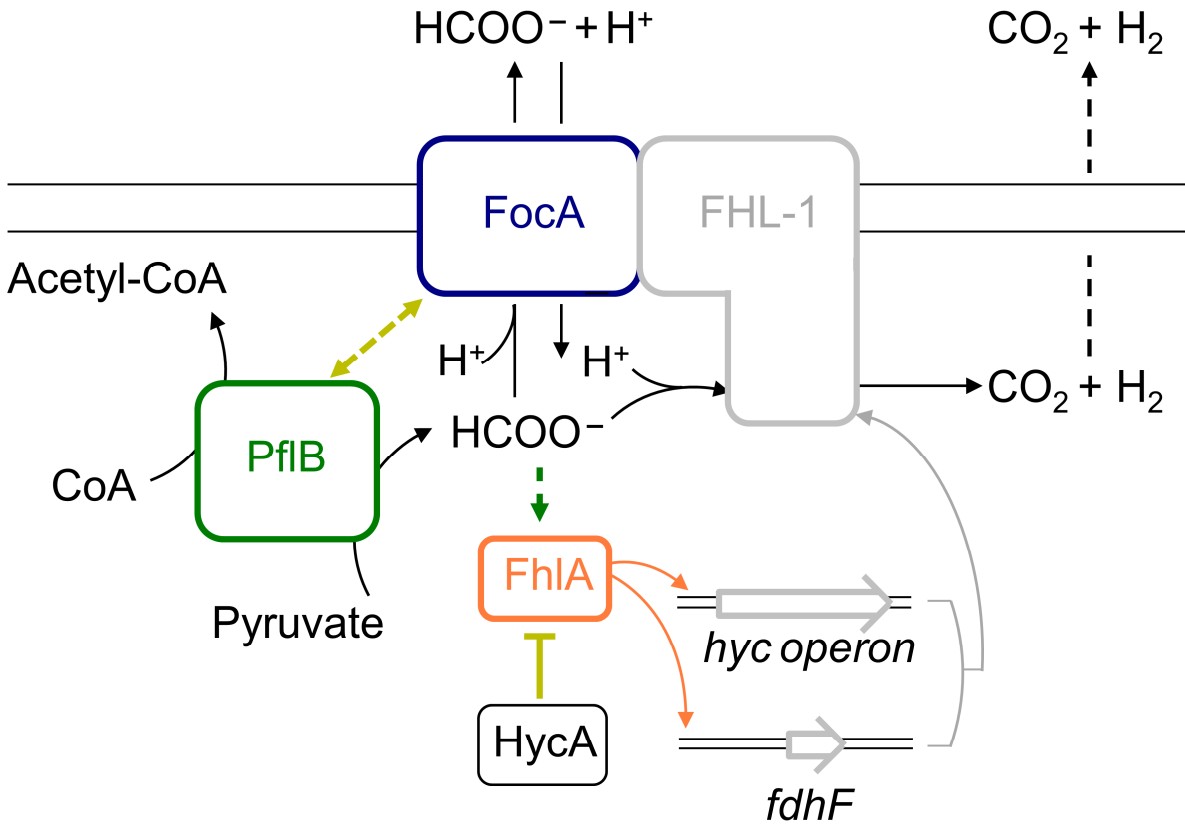

**Figure 4.** Model depicting how formate levels in *E. coli* are controlled during glucose fermentation. The cytoplasmic membrane is represented as parallel lines with the cytoplasm below the lines. Black straight or curved full lines represent conversions of metabolites; the dashed black line represents gaseous diffusion across the membrane; light green lines represent regulatory protein interactions, with the arrowhead indicating a positive effect and the bar a negative effect; the dashed dark green line represents formate signalling; and the orange arrows indicate transcriptional activation. Abbreviations: FHL-1, formate hydrogenlyase complex I; PflB, pyruvate formate-lyase.

Considering that PflB has a high catalytic turnover number and the enzyme's chemical equilibrium strongly favors pyruvate cleavage [8], the pyruvate generated during glycolysis in early exponential phase cells will generate formate rapidly, which necessitates an efficient means of formate removal from the cytoplasm. This likely explains why the introduction of a plasmid carrying the *focA-pflB* operon shows enhanced formate efflux with concomitantly lower intracellular levels of formate. More severe, or long-term, perturbations to the system, such as the introduction of mutationally altered FocA variants with significantly reduced (strain DH4200) or increased (strain DH4300) cytoplasmic formate levels, all result in a reduction in growth rate, particularly in the early stages of exponential growth, i.e., when glucose concentrations are high. Reduced growth rates resulting from changes in the equilibrium of intracellular versus extracellular formate are proposed to be the consequence of either decreased cytoplasmic pH (high intracellular formate), loss of formate as a valuable energy and electron source (low intracellular formate concentration), loss of a potential carbon source (low intracellular formate causing lower $CO_2$/bicarbonate levels), or through impaired energy conservation (too high or too low formate levels) [3]. Despite formate not being considered to act as a direct carbon source for *E. coli* [4,6], it is nevertheless an excellent electron donor, is a potential source of bicarbonate [36], and has an important function in maintaining pH homeostasis [3,6.20]. The growth deficiency of strain DH4200 can at least be partially phenotypically complemented by supplementation of amino acids to the growth medium [13], which might be due to a compensatory extracellular pH increase or via an indirect effect on intracellular bicarbonate levels [36]. A further possibility

is that the PflB-FocA-FHL-1 axis (Figure 4) contributes to energy conservation through *pmf* generation (discussed in [3,6]), whereby accumulation of cytoplasmic formate, or its too rapid and efficient efflux, may negatively affect the cell's ion gradients. Future studies will, therefore, focus on determining the cause(s) of the negative growth phenotype resulting from perturbations in FocA-dependent formate translocation, and how PflB and the FHL-1 complex contribute to the overall mechanism of maintaining FocA-dependent formate homeostasis.

**Author Contributions:** Conceptualization, M.K. and R.G.S.; Experimental analyses, M.K.; Data analysis, M.K. and R.G.S.; Writing, reviewing, and editing, M.K. and R.G.S.; Supervision, R.G.S.; Funding acquisition, R.G.S. All authors have read and agreed to the published version of the manuscript.

**Funding:** This research was funded by the Deutsche Forschungsgemeinschaft, grant number SA-494/11-1 and the Martin Luther University Halle-Wittenberg.

**Institutional Review Board Statement:** Not applicable for studies not involving humans or animals.

**Informed Consent Statement:** Not applicable for studies not involving humans.

**Data Availability Statement:** All data generated during the study are included in the paper.

**Conflicts of Interest:** The authors declare no conflict of interest.

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
