# Peer review of "Coordinated Expression of the Genes Encoding FocA and Pyruvate Formate-Lyase Is Important for Maintenance of Formate Homeostasis during Fermentative Growth of Escherichia coli"

_fermentation, doi:10.3390/fermentation9040382_

Round 1

Reviewer 1 Report

The manuscript by Kammel and Sawers deals with the role of FocA, a membrane channel facilitating formate/formic acid transport across the cytoplasmic membrane of E. coli, in the course of fermentation. A huge set of strains was combined with various plasmids expressing different variants of focA from the genome or plasmid-based and the respective intracellular and extracellular formate levels were compared. The authors conclude that co-localisation and co-expression are the ultimate determinants to ensure proper formate levels during fermentation.

The conclusion, however, goes a bit far since co-localisation was not assessed at all and the effects might also be explained by strong differences in the formate import or export efficiencies of the analysed variants. Either the discussion and abstract need to be revised or additional experiments using constructs expressing focA and PFL from one plasmid need to be carried out.

The authors furthermore neither mentioned nor considered the fact that strain DH701 encodes a focA gene that includes two stop codons in the middle of the gene. Since the GUG to AUG codon modification in DH702 shows polar effects on pflB, also strong effects in strain DH701 have to be expected. Therefore, the use of an isogenic focA deletion is highly recommended here.

In addition, the possible effects of formate disproportionation by FHL should be discussed more extensively. The FHL activity probably affects the cytoplasmic formate levels. Additional experiments in an FHL deletion strain background would be appropriate.

Specific comments:

A bit more care on correct formatting of the document should be taken. Italicisation of genes and strains is absent starting from the results section.

Table 1: The relevant strain characteristics should be mentioned here so that the reader does not need to go back to all the references.

Which start codon is used in case of the plasmid-encoded focA variants? Please indicate the promoter in order to enable the reader to at least assume the strength of gene expression.

Please check the units in line 101, i.e. µmol/ml is probably µg/ml.

Line 124: It would be helpful if strain DH701 would be described as “delta focA” rather than “focA”

Line 171: Is anything known about the functionality of the FocA expressed from the plasmid? Strong overexpression often leads to non-functional protein.

Line 178/179: “Introduction of pfocA had no effect on extracellular levels of formate compared with strain DH701 (Fig. 3b).” Why is the formate level in DH701 not lower than that of DH4100? This doesn’t fit with the result presented in Fig. 2a.

Line 191: “codon of chromosomal the focA” please correct

Legend to figure 3: Please provide an extended description of strain DH702. Due to the manuscript organization the reader will read the figure legend before the description on the GTG vs ATG start codons in the main text.

Lines 267-270: The statement needs to be weakened. No experiments have been included that directly analyse the effects of focA-pflB coupling by e.g. expression from a plasmid encoding both genes.

Line 292: “if” instead of “of”

Lines 304-306: The statement is based on the fact that the H209N variant cannot be rescued by expression of the wild-type focA gene in trans. The authors should consider that the same phenomenon might be observed if the export efficiency of FocA-H209N is so much higher compared to the import efficiency of wild-type focA. This needs to be included in the discussion or further experiments need to be carried out. In addition, strain DH4300(T91A) can be rescued by focA expression in trans to the level of DH702. This somehow sheds doubt on the clear conclusion that only correct expression from the same operon leads to balanced formate levels.

Line 329: “periplasmic enzymic sequestration of formate” – is this likely? Are the FDHs not mainly biased towards formate oxidation?

Reviewer 2 Report

The article under review is dedicated to a very important problem of fermentation, which is the regulation of intracellular acid homeostasis in E. coli, as a model organism performing mixed acid fermentation during glucose fermentation. The article is clearly and comprehensibly readable, interesting and new facts/data are obtained. During the production of acids, and especially formic acid, mainly the involvement of FocA, PflB and FHL-1 are discussed in regulation of intracellular level of formate is discussed. The effects on intracellular formate levels of over-producing different FocA variants in the wild-type strain and different focA mutants were examined:  a strong evidence supporting the previously suggested role of FocA in maintaining formate homeostasis in fermenting E. coli cells was stated.

I would suggest that the authors specify the conditions for the proposed model (Fig. 4) in the discussion and conclusion, for example, pH, carbon source, etc., since another channel (FocB) is known as formic acid channel, which also has a role in intracellular formate concentration regulation, the existence of what is not introduced in the article. 
